# Unlocking Cellular Memory and Gene Regulatory Networks: Pioneering the Future of Therapeutic Innovations

**DOI:** 10.3390/cells14120903

**Published:** 2025-06-14

**Authors:** Md Sorique Aziz Momin, Jhuma Bhadra, Debmalya Bhunia, Achinta Sannigrahi, Nayan De

**Affiliations:** 1School of Physics and Astronomy, College of Science, Rochester Institute of Technology, Rochester, NY 14623, USA; sxasps@rit.edu; 2Department of Chemistry, Sarojini Naidu College for Women, Kolkata 700028, India; jbhadra.chem@sncwgs.ac.in; 3Cold Spring Harbor Laboratory, 1 Bungtown Rd, Cold Spring Harbor, NY 11724, USA; bhunia@cshl.edu; 4Department of Molecular Genetics, University of Texas Southwestern Medical Center, 5323 Harry Hines Blvd, Dallas, TX 75390, USA; 5Institute for System Biology, 401 Terry Ave N, Seattle, WA 98109, USA

**Keywords:** gene regulatory networks, cellular memory, noise, information theory, drug resistance

## Abstract

Cellular memory is the competence of cells to preserve information from past experiences and respond aptly. This memory is maintained and controlled by gene regulatory networks (GRNs). GRNs are crucial for understanding why some cells are resistant to treatment, particularly for cancer. In our study, we created a new mathematical model to understand how “noise” affects cellular memory in GRNs, focusing on a “double positive feedback loop”. Our theoretical perspective article equipped with mathematical modeling exhibits how noise and feedback loops interact in GRNs. It also proposes a potential theoretical avenue for future therapy. By targeting the mechanisms that maintain drug resistance in cells, we aim to develop therapies that can restore the sensitivity of cancer cells to treatment.

## 1. Introduction

Cellular memory is an invisible thread that holds together an individual’s experience through their own genetic arrangements. However, disorder in genetic structures leads to disease, when memory fails to protect, leading to chaos rather than harmony in the genetic regulatory system. Retaining cellular memory after multiple rounds of cell division is essential for ensuring proper tissue/organ function, and also for coordination of complex cellular processes [1,2]. Cellular memory is pivotal to the organism for the preservation and survival of these expression and to maintain functionality over its lifespan [3].

Cellular memory lies at the transcriptional state of a cell [4]. It operates through bistable configurations which alters between active (“on”) and inactive (“off”) modes, ensuring essential gene expression [5]. This phenomenon is particularly relevant for diseases like cancer, where cellular memory contributes to key characteristics, such as drug resistance. Abnormal genetically expressed cells are prone to switch between a drug-susceptible state and a drug-resistant state. The ability of genetically identical cells to exhibit distinct behaviors, known as cellular heterogeneity, provides another layer of complexity to cellular memory. This heterogeneity often arises from non-genetic transcriptional variations under identical conditions [6,7,8,9,10,11,12]. These non-genetic differences create subpopulations of cells capable of evading treatment, leading to relapse in patients [13]. Among these, rare pre-resistant cells stand out as critical drivers of drug resistance. These cells display temporary features due to upregulation of multiple marker genes. Hence, potential approaches need to be considered for disrupting drug resistance [14]. Therefore, cellular intermediate memories suffer through multiple cell divisions, generating vulnerability to transitions and potential therapeutic approaches needed to maintain “cellular memory” [1,15].

A gene regulatory network (GRN) is a collection of genes which interact to control genetic expression levels within a cell. It provides a powerful framework for understanding cellular memory and regulatory mechanisms [16,17]. GRNs play a central role in sustaining transcriptional memory, as their feedback loops stabilize gene expression patterns over time [18]. Positive autoregulation helps lock genes into active states, ensuring stability once a transcriptional state is established. Double positive feedback loops—where two genes mutually enhance each other′s expression—are especially critical for maintaining bistable gene expression states, reinforcing the memory required for consistent cell functions [19]. However, GRNs are susceptible to disruption; random fluctuations in gene expression, or noise, accumulate over successive cell divisions and can destabilize the mutual reinforcement within feedback loops [20]. This destabilization jeopardizes transcriptional memory, potentially leading to a loss of cellular identity, as seen in pathological conditions such as cancer, where disrupted memory states contribute to malignancy [21,22]. Despite these challenges, advances in cellular and synthetic biology offer promising strategies for manipulating cellular memory. Emerging tools in cellular reprogramming allow for precise modifications of transcriptional states, enabling the conversion of specialized cells into pluripotent stem cells or other functional types, paving the way for personalized regenerative medicine [23,24]. Clustered regularly interspaced short palindromic repeats (CRISPR)-based genetic and epigenetic editing technologies further enhance our ability to dynamically modulate GRNs, restoring healthy transcriptional states and targeting diseases caused by disrupted networks, such as cancer and neurodegenerative disorders [25,26]. Synthetic biology enables the design of synthetic memory circuits that program cellular behaviors with high precision, facilitating biological computing and targeted therapeutic interventions. By reprogramming cellular memory on demand, synthetic biology holds promise for addressing diseases like cancer, where pathological memory states contribute to drug resistance and therapeutic failure [27,28,29]. Small molecule inhibitors can disrupt the signaling pathways that stabilize aberrant memory states, prompting cells to revert to healthier configurations, while peptide nucleic acids (PNAs) offer potential strategies for modulating the DNA-binding regions of master regulators within GRNs to alter gene expression and force transitions in cellular memory states [30,31].

This study explores the implications of manipulating cellular memory and GRNs, particularly in modulating drug susceptibility and resistance in cells (Figure 1). Cellular memory, governed by gene expression dynamics, plays a crucial role in determining how cells respond to therapeutic agents. By analyzing transcriptional bursting—episodic and stochastic fluctuations in gene expression—this research aims to uncover the variability in cellular responses and to identify strategies for better controlling these dynamics. A key aspect of this analysis is the mutual information shared between genes within GRNs, which quantifies the dependency and communication between their expression states. Noise in gene expression [32], arising from both extrinsic sources (such as variations in cellular components like transcription factors) and intrinsic sources [33] (stemming from the stochastic nature of transcription and translation), significantly impacts this mutual information [34]. High levels of noise can weaken communication within GRNs, disrupting gene coordination, which is essential for maintaining stable cellular memory states. Understanding the interplay between noise [35] and mutual information is crucial for developing therapeutic interventions aimed at restoring or reprogramming cellular memory. However, in our theoretical perspective article, which is a combination of unique mathematical modeling frameworks and advanced therapeutic strategies (involving inhibitors, CRISPR tools, and synthetic biology), we approach the interconnected reprogramming of cellular memory to overcome drug resistance and to improve drug efficacy.

## 2. Cellular Memory and GRNs in Drug Resistance and Susceptibility

Cell state transitions, which result in the loss of prior cellular memory, can take place without direct genetic modifications. Shaffer et al. pioneered a lineage-tracing approach to assess gene expression memory and identify cells undergoing such state changes (Figure 2C). This technique, termed scMemorySeq [36], integrates single-cell RNA sequencing (scRNA-seq) with cellular lineage barcoding to analyze the persistence of gene expression states at a single-cell resolution [37]. Utilizing melanoma cells as a model, Bai et al. monitored cell lineages transitioning from a drug-sensitive state to a state predisposed to drug resistance [38]. Their investigation revealed that the TGF-β and PI3K pathways play pivotal roles in regulating these transitions (Figure 2B). A significant takeaway from this study is that modulating cell state transitions can substantially mitigate drug resistance. Interestingly, altering specific signaling pathways can globally reconfigure gene expression states, thereby influencing drug susceptibility [39]. For example, preconditioning cells with a PI3K inhibitor (PI3Ki) shifted them into a MAPK-dependent transcriptional state, thereby enhancing their sensitivity to MAPK inhibitors. While previous studies have explored PI3Ki′s role in overcoming melanoma resistance, this research demonstrated that even a short pretreatment with PI3Ki before targeted therapy substantially lowers resistance [40]. This finding introduces a potential strategy to enhance drug sensitivity across heterogeneous cell populations by transiently modifying gene expression states.

In the context of cancer, cellular memory over intermediate timescales has been associated with critical characteristics such as drug resistance. These memories, although enduring across multiple cell divisions, remain reversible, making them vulnerable to state transitions. Cells generally exist in either a drug-susceptible state or a drug-resistant state and can dynamically transition between these states (Figure 2A) [15,41]. Shaffer et al. demonstrated that untreated melanoma cells inherently fluctuate between these states, existing in both drug-susceptible and primed conditions. To uncover the molecular regulators governing these transitions, the researchers employed scMemorySeq, a combination of cellular barcoding and scRNA-seq, enabling the mapping of heritable gene expression states and their shifts over time. Cellular barcoding facilitates high-throughput lineage tracking, while scRNA-seq deciphers transcriptional states at a single-cell level [42]. Through analyzing gene expression within the same lineage, they inferred cellular memory. When cellular memory persists, all descendant cells maintain the same transcriptional state as the initial cell. By contrast, if memory is lost, lineages diversify, giving rise to distinct gene expression profiles [15]. Applying scMemorySeq to BRAF V600E-mutated WM989 melanoma cells, the researchers employed a barcode library encompassing both drug-susceptible and primed states [43,44]. Their study, involving 12,531 melanoma cells (with 7581 barcode-labeled cells), identified two primary transcriptionally distinct populations [15]. One cluster exhibited high expression of primed-state markers, including EGFR and AXL, whereas another population predominantly expressed drug-susceptible genes, such as SOX10 and MITF. These distinct states were consistently observed using Louvain clustering and other dimensionality reduction techniques.

Cell signaling pathways (Table 1) play a crucial role in driving the transition between drug-susceptible and drug-resistant states. Notably, the TGF-β signaling pathway facilitates the shift from the drug-susceptible state to the primed state. To validate this, Shaffer et al. introduced a high-complexity transcribed barcode library into WM989 cells [13]. Using Louvain clustering, they classified cells as either primed or drug-susceptible, with most primed-state cells expressing known marker genes. Their findings demonstrated that exposure to TGFB1 increased the proportion of primed-state cells, marked by the upregulation of genes, such as NGFR, FGFR1, FOSL1, and JUN [45]. Conversely, transitioning cells back to a drug-susceptible state was achieved using the PI3Ki treatment [46], which led to a 93% reduction in primed-state cells across lineages. These findings collectively highlight that both TGFB1 and PI3Ki actively drive state transitions. Importantly, the observed changes in primed-state cell numbers were attributed to state switching rather than broader population dynamics. This research underscores the potential of leveraging transient signaling modulation to influence gene expression states and to enhance therapeutic susceptibility. By targeting key signaling pathways, clinicians may develop adaptive treatment strategies to counteract drug resistance and to improve patient outcomes for melanoma and potentially for other cancers [47].

## 3. Mathematical Modeling of Noise Dynamics and Mutual Information in Gene Regulation

In this section, we develop mathematical models [57,58] to describe the key mechanisms of gene regulation underlying cellular memory [59,60,61]. These models focus on capturing essential regulatory motifs, such as the double positive feedback loop (Figure 3), which plays a crucial role in maintaining stable gene expression states [62,63]. Additionally, we explore the impact of gene regulation noise on information processing by deriving analytical expressions for variances, covariances, and mutual information. These metrics quantify the interplay between feedback mechanisms, noise, and the stability of cellular memory.

A GRN motif can be described using the concentration or copy number of its gene products. For a gene product X, synthesized at a constant rate $k_{x}$, and degraded at rate $\mu_{x}$, the dynamics are as follows: $\frac{dX}{dt}=k_{x}-\mu_{x}X$. When a transcription factor (TF) X activates production of a protein Y, it binds DNA D to form a complex $\left[XD\right]$, from which Y is produced and later degraded. The reaction dynamics are as follows: $X+D⇌\;\left[XD\right]\to X+D+Y$, Y→∅. Assuming rapid TF-DNA binding relative to production, $\left[XD\right]$ reaches a quasi-steady state. The fraction of DNA bound is approximated as $\frac{\left[XD\right]}{D_{T}}=\frac{X}{K_{xy}+X}$, where $K_{xy}=k_{off}/k_{on}$. Substituting this into the equation for Y, we obtain the following: $\frac{dY}{dt}=k_{xy}\frac{X}{K_{xy}+X}-\mu_{y}Y$. This is a Hill-type function where $K_{xy}$ indicates the TF concentration needed for half-maximal Y production. Lower $K_{xy}$ values imply higher promoter affinity and stronger activation (see the Appendix A for the detailed derivations).

The double positive feedback loop is a fundamental and widely studied motif in gene regulation. In this system, two genes, *X* and *Y*, mutually enhance each other’s production, creating a robust feedback mechanism. The dynamics of this system are described by the following coupled ordinary differential equations [62,64,65]:(1)$$\frac{dX}{dt}=k_{xy}\frac{Y^{n}}{K_{xy}^{n}+Y^{n}}-\mu_{x}X+\xi_{x}\left(t\right)$$(2)$$\frac{dY}{dt}=k_{yx}\frac{X^{n}}{K_{yx}^{n}+X^{n}}-\mu_{y}Y+\xi_{y}\left(t\right)$$

Here, *X* enhances the production of *Y*, and *Y* reciprocally enhances the production of *X*. The synthesis rate constants $k_{xy}$ and $k_{yx}$ represent the rates of production for *X* and *Y*, respectively. The Hill functions $\frac{Y^{n}}{K_{xy}^{n}+Y^{n}}$ and $\frac{X^{n}}{K_{yx}^{n}+X^{n}}$ describe the cooperative nature of this feedback [66,67]. For the Hill function $\frac{Y^{n}}{K_{xy}^{n}+Y^{n}}$, *Y* represents the concentration of the protein enhancing *X*, *n* is the Hill coefficient indicating cooperativity, and $K_{xy}$ is the half-maximal effective concentration of *Y* required to achieve half the maximum production of *X*. A higher *n* reflects stronger cooperativity, while a lower $K_{xy}$ indicates greater sensitivity to *Y*. The sigmoidal shape of the Hill function ensures that the production of *X* is low at low *Y* concentrations, but increases rapidly as *Y* exceeds $K_{xy}$. Similarly, $\frac{X^{n}}{K_{yx}^{n}+X^{n}}$ describes how *X* enhances the production of *Y*, with $K_{yx}$ and *n* playing analogous roles. The terms $\mu_{x}X$ and $\mu_{y}Y$ represent the degradation rates of *X* and *Y*, respectively [67]. Together, these interactions form a bistable feedback system, enabling the system to stably exist in the high or low expression states of both *X* and *Y*. This biostability underpins cellular memory and allows cells to maintain distinct gene expression patterns in response to environmental or developmental signals [5,67].

The feedback loop is also subject to stochastic fluctuations, commonly referred to as gene regulation noise. Noise originates from the random binding and unbinding of transcription factors to gene promoters, resulting in variability in gene expression levels. These fluctuations in transcriptional activity play a crucial role in determining the stability and reliability of gene expression states. The random binding events are governed by factors such as the binding affinity of transcription factors, their concentration, and the kinetics of the binding process. As transcription factors randomly interact with the promoter, the rate of gene expression fluctuates, leading to variability in the levels of the protein products. Understanding the impact of such noise is essential for analyzing how cells process information and sustain stable expression states despite the inherent randomness. To quantify the effects of feedback and noise on cellular memory, we derive analytical expressions for the variances and covariances of gene expression in the steady state. The noise terms $\xi_{x}\left(t\right)$ and $\xi_{y}\left(t\right)$ are modeled as independent white Gaussian noise, with zero mean and a noise correlation function as follows [68]:(3)$$\langle \xi_{i}\left(t\right)\xi_{j}\left(t^{\prime }\right)\rangle =\langle {\left|\xi_{i}\right|}^{2}\rangle \delta_{ij}\delta \left(t-t^{\prime }\right)$$

By linearizing the system around the steady state [69,70,71] and considering small perturbations from the mean values of *X* and *Y*, we solve the following Lyapunov equation [72,73]:(4)$$J\Sigma +\Sigma J^{T}+D=0$$

In this equation, *J* represents the Jacobian matrix of the system, Σ is the (co)variance matrix, and *D* is the noise matrix. Solving this equation allows us to obtain the variances Σ(*X*), Σ(*Y*), and the covariance Σ (*X*, *Y*) (see the Appendix A for detailed analytical calculations). These quantities are critical for calculating the mutual information between *X* and *Y*. Mutual information (MI) quantifies how much knowing the value of one gene product reduces uncertainty about the other, capturing how closely two variables, *X* and *Y*, are related. When *X* and *Y* are strongly connected, MI is high, meaning that knowing *X* gives substantial insight into *Y*; conversely, if *X* and *Y* are unrelated, MI is zero, indicating that *X* provides no useful information about *Y*. For Gaussian random variables, the mutual information between *X* and *Y* is given by the following equation [62,74]:(5)$$I\left(X;Y\right)=\frac{1}{2}{log}_{2}\left[\frac{\Sigma \left(X\right)}{\Sigma \left(X\left|Y\right.\right)}\right]$$

This expression offers a clear measure of how feedback loops and noise affect the shared information between gene products. MI thus reveals how noise and feedback influence information flow in gene regulatory networks, providing insights into cellular memory mechanisms. By examining these relationships, we better understand how cells maintain stable gene expression patterns and consistently transmit information, even in the presence of stochastic fluctuations. While our current model focuses on a two-node double positive feedback loop for conceptual clarity, it is readily extendable to more complex gene regulatory networks (GRNs) involving additional nodes and mixed feedback motifs. Many GRNs contain diverse topologies—such as autoregulatory, double positive, and double negative loops—known to support bistable or multistable behavior [17].

We begin by considering a basal gene expression level of *X* = 5 copies, where gene *X* acts as a transcription factor that positively regulates the expression of gene *Y* by increasing the synthesis rate parameter $k_{yx}$. This, in turn, enhances the synthesis rate of protein *Y* (*X* → *Y*). The newly synthesized protein *Y* then acts as a transcription factor for gene *X*, further boosting the synthesis rate of protein *X* (*Y* → *X*). This reciprocal regulation forms an interlocked positive feedback loop, where the expression levels of both genes continuously influence each other. The system operates in a bistable manner, where one steady state corresponds to both genes being ON, and another corresponds to both genes being OFF. A signal that activates either protein *X* or *Y* can drive the system into the ON state, effectively “locking” the genes in a high-expression state, supporting cellular memory.

The expression levels of genes *X* and *Y* are modulated by adjusting the rate parameters $k_{yx}$ and $k_{xy}$, which influence the activity of RNA polymerase and the mRNA production rate per unit time. By increasing these parameters, the gene expression levels of both *X* and *Y* are increased. The nonlinear interactions between these genes, amplified by the cooperative binding of transcription factors to their respective promoter regions, generate noise in the system. As gene expression increases, it requires more transcription factors to activate the genes, which results in a higher demand for these molecules. This increased demand for transcription factors, combined with the inherent stochastic nature of their availability and binding, introduces further noise into the system. The more complex the regulation, the greater the fluctuations in gene expression, further contributing to the overall instability and unpredictability of the system.

The increase in gene expression variability is evident in the rise of variance and covariance of genes *X* and *Y*. As shown in Figure 4A, the variance of *X* increases with the expression level, indicating higher fluctuations in its copy number. Similarly, the variance of *Y*, depicted in Figure 4B, follows the same trend, confirming that both genes experience greater expression variability under high-expression conditions. The covariance between genes *X* and *Y*, shown in Figure 4C, also increases, further illustrating the growing instability in gene expression. In a well-regulated feedback loop, the coordinated activation of the two genes ensures a stable and predictable expression pattern. However, as stochastic fluctuations become more pronounced, they weaken the stability of this regulation. This loss of coherence is further quantified by a decline in mutual information, as shown in Figure 4D, which measures the degree to which the expression level of one gene can reliably predict the expression level of the other. When mutual information is high, gene *X* and gene *Y* are tightly coordinated, reinforcing each other’s expression in a stable manner. As mutual information decreases, their correlation weakens, making gene expression increasingly erratic. This loss of mutual information reflects the progressive breakdown of the feedback mechanism, ultimately impairing the ability of the system to retain the memory of prior gene expression states. The increase in stochastic noise and the reduction in mutual information thus contribute to the destabilization of the high-expression state, making it more susceptible to random shifts between active and inactive states.

This disruption in gene regulation has direct consequences for cellular phenotypes, particularly in the context of drug resistance. In many biological systems, drug resistance is governed by a bistable regulatory network, where one state corresponds to drug susceptibility and the other to resistance. The double positive feedback loop ensures that, once resistance-associated genes are activated, they remain expressed, maintaining the resistant state even after drug removal. However, as noise increases and mutual information decreases, the ability of the system to sustain the resistant state is compromised. The reduction in mutual information implies that the regulatory connection between the two genes becomes weaker, making the resistant state more prone to stochastic transitions. As a result, cells that were previously locked in a resistant state can spontaneously revert to a drug-susceptible state. This phenomenon could explain the heterogeneous response observed in populations where some cells lose resistance despite previous adaptation to drug exposure [77,78,79].

These findings suggest that high levels of gene expression, while reinforcing the resistant state under stable conditions, can ultimately introduce excessive fluctuations that destabilize this state [80]. This increased variability enhances the probability of resistance loss, leading to re-sensitization to drug treatment. From a therapeutic perspective, this highlights the potential for interventions that amplify gene expression noise or disrupt the coordination of resistance-associated genes. By targeting the regulatory instability caused by excessive gene expression, it may be possible to drive resistant cells back into a drug-susceptible state, thereby improving the effectiveness of drug treatments.

This work integrates a classical gene regulatory motif with an information-theoretic framework to quantitatively explain how gene expression noise impacts mutual information and cellular memory, particularly in drug resistance. While it is known that higher expression can increase noise, we go beyond this intuition by deriving analytical expressions for variance, covariance, and mutual information using stochastic modeling and Lyapunov analysis. We explicitly quantify how increasing expression strength degrades mutual information, a rarely addressed aspect in models of bistability, and bridge information theory with gene regulatory dynamics to understand how feedback and noise affect epigenetic state stability. Contextualized by experimental data from Shaffer et al. on drug-resistant melanoma, our model provides a mechanistic explanation for reversible resistance: an elevated expression initially stabilizes resistance but ultimately amplifies noise, eroding memory. This framework not only deepens theoretical insight but suggests that targeting gene expression noise or mutual information could destabilize resistant states, offering a potential strategy for therapeutic re-sensitization.

## 4. Inhibition of Cellular Memory with Selective Inhibitor

Selective inhibitors are powerful tools for disrupting cellular memory by targeting the pathways that sustain specific cellular states. Researchers are exploring strategies to induce state transitions, which may reset cellular behavior and enhance therapeutic efficacy. A study by Shaffer et al. tested a sequential dosing approach on melanoma cells, where cells were first pretreated with state modulators, such as TGFB1 or PI3K inhibitors (PI3Ki), followed by treatment with targeted therapies, like BRAFi/MEKi. For the TGFB1 pretreatment, the strategy induced state switching, leading to a more rapid elimination of drug-sensitive cells while reducing the development of resistant populations after BRAFi/MEKi therapy. However, the priming effect of the TGFB1 pretreatment was less pronounced compared to the use of TGFBR inhibitors (TGFBRi). When PI3Ki was used as a pretreatment, it resulted in a 62% decrease in resistant colonies and a 57% reduction in resistant cells compared to BRAFi/MEKi alone. Additionally, combining PI3Ki with BRAFi/MEKi was highly effective, eliminating almost all resistant cells. However, this combination treatment caused significant systemic toxicity, suggesting the need for lower doses in clinical trials to minimize adverse effects.

Shaffer et al. also tested a pretreatment regimen involving TGFB1 and PI3Ki prior to BRAFi/MEKi administration. In this approach, cells were pretreated for five days before receiving four weeks of BRAFi/MEKi therapy. The TGFB1 pretreatment led to a faster elimination of drug-susceptible cells compared to BRAFi/MEKi alone, and resulted in fewer resistant cells at the end of the treatment period. By contrast, pretreatment with TGFBRi had a minimal impact on resistance, indicating it did not significantly alter the progression of drug resistance. The PI3Ki pretreatment reduced the number of resistant cells and colonies, further supporting its potential as an effective strategy. However, the combination of PI3Ki with targeted therapy led to toxicity, highlighting the necessity for clinical trials that explore lower dosages to reduce side effects.

## 5. CRISPR’s Role in “Reverse Drug Resistance”

CRISPR is an advanced technology for genetic modification [81]. Basically, it is a powerful gene-editing tool which allows for precise modifications to the DNA of living organisms [82]. Due to specific genetic expression, cells that are susceptible to drugs become drug-resistant in some portions [15]. When cells switch states (from drug-susceptible to drug-resistant), they lose memory of their prior state, and alter the drug effectivity. However, in this direction, CRISPR can potentially inhibit genetic switching related to drug resistance in various ways (Figure 5).

The impact of drug resistance is one of the main obstacles in the treatment of cancer. Mutations in some genes of different cellular signaling pathways are associated with drug resistance. CRISPR can target these genes and knock down those specific genes which are responsible for cellular drug resistance, making the cells susceptible to the drug again [83,84]. Moreover, the CRISPR/Cas9 system might revert resistance to gene mutations and identify potential resistance targets in drug-resistant breast cancer [78].

CRISPR can target the regulatory genes which control expression of resistance factors. By altering the regulation of certain genes, CRISPR can prevent the cells either from switching the resistance state or promote the loss of resistance [85,86]. Moreover, CRISPR may combine with other therapeutic strategies (PNAs, antibiotics, or targeted cancer therapy) for higher efficiency to reverse drug resistance. It will involve not only targeting resistance genes but enhancing the effectiveness of the drugs by editing genes [87]. CRISPR with fluorescent marker may be applied for real-time monitoring of specific epigenetic modifications for “reverse drug resistance” in living cells. It may help to track the genetic regulation after different inducements through the CRISPR treatment during genetic state switching [83,84,85].

## 6. Cellular Memory for Next-Generation Therapeutic Breakthroughs

Synthetic biologists have envisioned what may be possible once we can reliably and predictably re-engineer biology. While building novel genetic circuits both in vitro and in vivo has been a pursuit of synthetic biologists for several years, most of these have yet to find utility in real-world applications. Nevertheless, these early efforts have proven useful both as research tools and in gaining a better understanding of natural biological mechanisms.

Synthetic biology is advancing the design of genetic devices that enable the study of cellular and molecular biology in mammalian cells. These genetic devices use diverse regulatory mechanisms to both examine cellular processes and achieve precise and dynamic control of cellular phenotype. Synthetic biology tools provide novel functionality to complement the examination of natural cell systems, including engineered molecules with specific activities and model systems that mimic complex regulatory processes. Cellular memory is responsible for the autoregulatory feedback loop present in the gene regulatory network in cellular systems. Negative feedback loops can reduce the noise in a biological system, and increase the responsiveness of the system to changes [67]. Using a set of synthetic transcriptional repressors, researchers compared the expression noise from a simple negative regulation system composed of a dox-inducible LacI repressor to that from a system in which autoregulatory negative feedback of the repressor was implemented via transcriptional modulation. The system encoding the negative feedback loop displayed significantly reduced levels of total transcriptional noise in mammalian cells, whereas the negative regulation system increased the intrinsic transcriptional noise. The above study indicates that a synthetically incorporated autoregulatory negative feedback loop in place of an autoregulatory feedback loop can rupture cellular memory.

Natural receptors, which detect specific endogenous inputs, can be engineered to generate a non-native output response. There are several examples of a native receptor being redirected to elicit a novel transcriptional response. One such approach exploits the modular structure of the receptor protein notch in the delta notch signaling pathway. In separate studies, both Struhl and Adachi and Sprinzak et al. showed that this notch receptor transcription factor module can be replaced by a synthetic transcription factor (Gal4-AD) so that, when activated in vivo, this chimeric notch receptor can activate genes targeted by the new transcription factor [88,89]. We can introduce the synthetic circuit to modulate cell signaling pathway and to stop the switching behavior of the cell from drug susceptible to a primed state (Figure 6).

## 7. Conclusions

Herein, we present a potential mathematical model based on an information–theoretical framework. It precisely measures noise within a biological network to interpret cellular memory in the context of drug resistance in cancer cells. Our approach revises the concept of transcriptional bursting in gene networks with double positive feedback loops, establishing a connection between mutual information and biological noise. This proposed model offers a clear depiction for the shifting of stable drug-resistant state transitions into a drug-susceptible state.

In addition, we proposed several strategies for disrupting cellular memory and overcoming drug resistance. These include cell signaling inhibitors, CRISPR technology, and synthetic biology. These selective inhibitors play a crucial role in disrupting cellular memory by targeting the pathways that maintain specific cellular states, offering the potential to reset cellular behavior and improve therapeutic outcomes.

This mathematical model, combined with both theoretical and simulated results, provides valuable insights into designing drugs that can disrupt cellular memory. We believe our perspective can open new avenues for researchers to address the current challenges in overcoming drug resistance in cells. However, the therapeutic implications of our proposed model for stabilizing cellular states in cancer and genetic disorders require experimental validation and precise modulation of gene regulatory networks (GRNs) across diverse cellular contexts.

## Figures and Tables

**Figure 1 cells-14-00903-f001:**
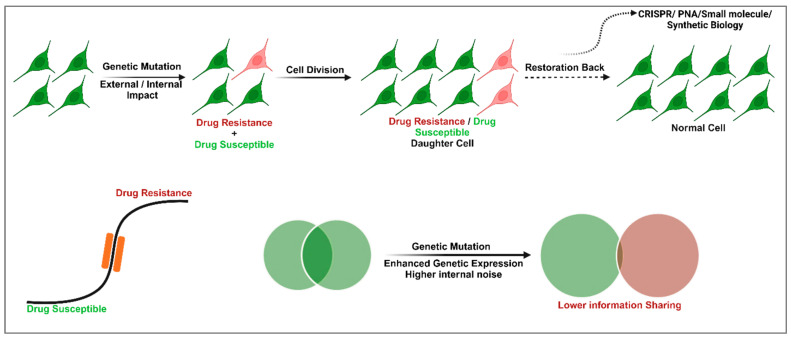
Overview of cellular memory, gene regulatory networks, and potential therapeutic strategies. The diagram illustrates the transition of normal cells into mixed populations of drug-susceptible and drug-resistant cells due to external or internal factors, including genetic mutations. Drug-resistant and drug-susceptible daughter cells arise through cell division. Using CRISPR, PNA, or small molecule-based synthetic biology approaches, memory can be restored, converting resistant cells back to their original state. The lower section highlights the progression from Stage 1 (drug-susceptible) to Stage 2 (drug-resistant), with potential blocking strategies to halt this transition. Genetic mutations lead to increased genetic expression variability and internal noise, disrupting mutual information sharing among cells. Restoring memory offers a path to therapeutic intervention.

**Figure 2 cells-14-00903-f002:**
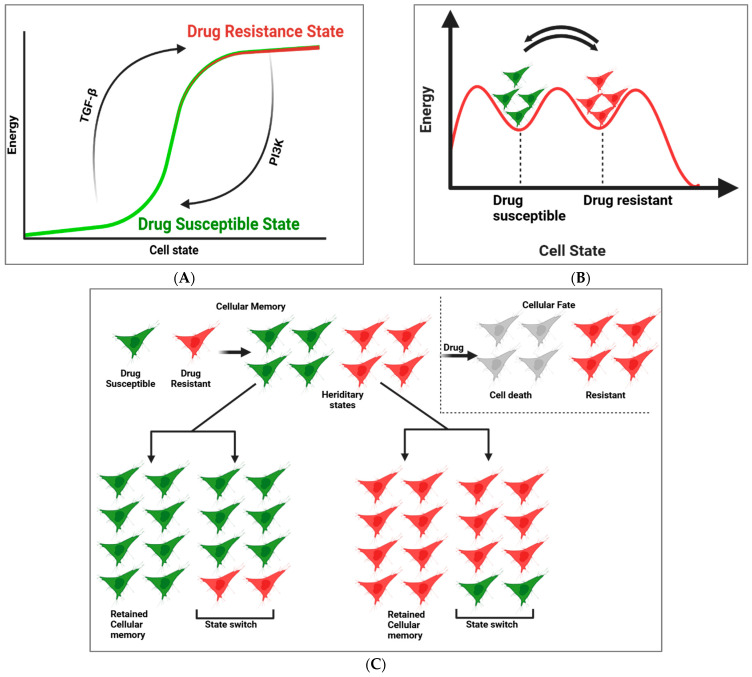
(**A**) Graph depicting cell state transitions between drug-susceptible and drug-resistant states. The green curve represents cells in a drug-susceptible state, while the red curve denotes cells in a drug-resistant state. Arrows indicate dynamic transitions between these states, illustrating the reversible nature of cellular states in response to external stimuli or treatment. (**B**) Represents a conceptual energy landscape model illustrating the relationship between the cellular energy state and drug susceptibility. The graph depicts two distinct energy minima, representing the drug-susceptible state (green cell cluster) and the drug-resistant state (red cell cluster), separated by an energy barrier (red line). The height of this barrier reflects the energy required for cells to transition between these states, thus influencing the stability of each phenotype. (**C**) Schematic illustrating lineage dynamics and memory retention. Green and red cells represent distinct cell states (e.g., drug-susceptible and drug-resistant). Lineages maintaining memory show consistent end states (green or red) across all progenies, while lineages losing memory exhibit mixed populations. The gray cells denote transient intermediate states during state transitions. Arrows depict lineage progression and state switching over time.

**Figure 3 cells-14-00903-f003:**
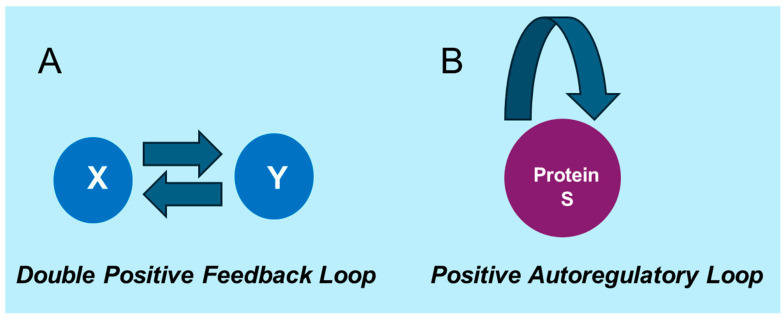
Graphical representation of (**A**) double positive feedback loop, and (**B**) positive autoregulatory loop networks in the biological system.

**Figure 4 cells-14-00903-f004:**
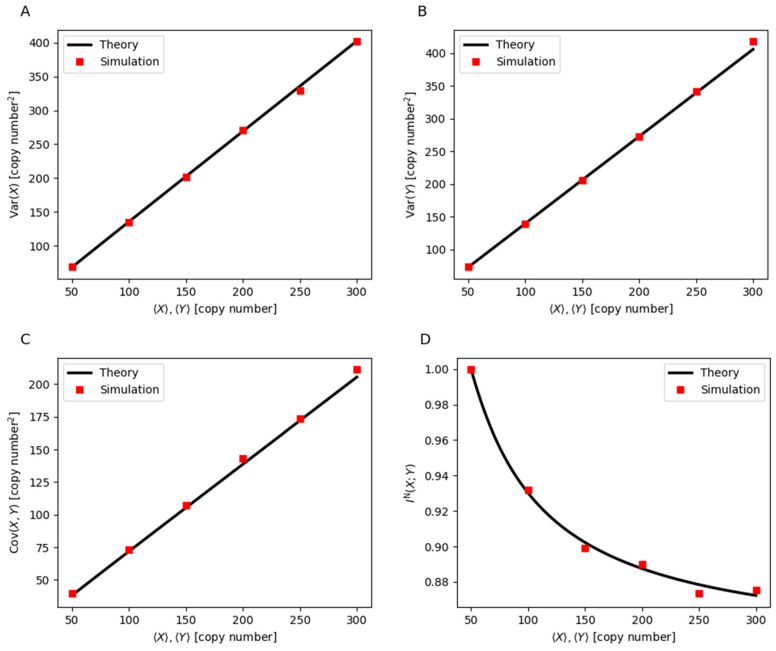
(**A**) The variance of *X*, (**B**) the variance of *Y*, (**C**) the covariance between *X* and *Y*, and (**D**) the normalized mutual information between *X* and *Y* as the gene expression level increases. The parameters used include n=1,${\;k}_{xy}=k_{yx}=5\;{min}^{-1}$, and $\mu_{x}=\mu_{y}=0.5\;{min}^{-1}$. Additionally, ${\;K}_{xy}=K_{yx}=\langle X\rangle =\langle X\rangle$, where $\langle \ldots \rangle$ denotes the mean copy number of the gene product. Analytical results are represented by lines, while results from the Gillespie stochastic simulation analysis [75,76] are depicted with symbols.

**Figure 5 cells-14-00903-f005:**
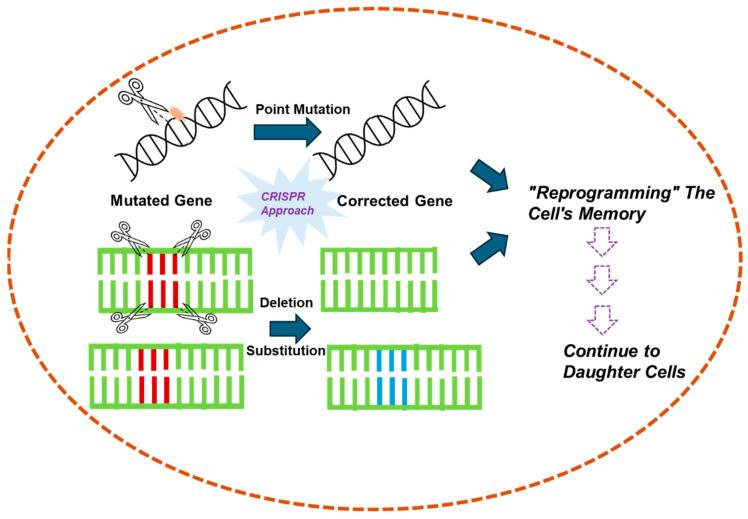
CRISPR-based genetic mutation elimination for restoration of cellular function against genetic diseases. The CRISPR-Cas9 gene-editing mechanism for targeted genomic modification. The process begins with guide RNAs (gRNAs) directing the Cas9 nuclease to a specific DNA sequence (highlighted in red), where Cas9 induces double-strand breaks (DSBs). These breaks activate cellular repair pathways, either through non-homologous end joining (NHEJ), which introduces insertions or deletions (indels) to disrupt the target gene, or homology-directed repair (HDR), which uses a repair template (highlighted in blue) to facilitate precise genetic modifications. The resulting edits alter gene expression, leading to functional changes in the cell, as depicted by the cascading arrows. The orange dotted line represents the cellular boundary, emphasizing that these processes occur within the cell.

**Figure 6 cells-14-00903-f006:**
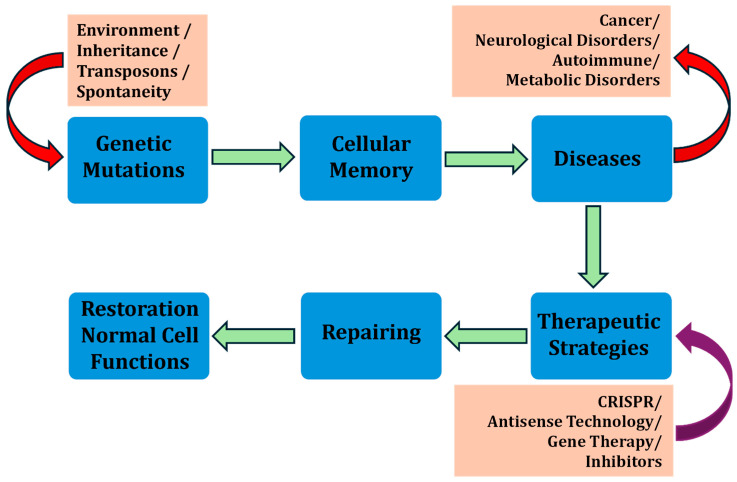
Schematic diagram for disease therapy based on shattering the gene regulatory network, and mutated cellular memory.

**Table 1 cells-14-00903-t001:** An overview of the key signaling pathways involved in cancer, along with their respective inhibitors in targeted cancer types.

Signaling Pathways	Inhibitors	Chemical Structure of Inhibitors
***TGF-β and PI3K signaling pathways*** are key regulators of melanoma cell survival, proliferation, and metastasis; targeting them may disrupt cancer adaptation and resistance.	BRAFi (e.g., Vemurafenib) and MEKi (e.g., Trametinib) target the MAPK pathway, commonly mutated in melanoma, and work synergistically to inhibit tumor-promoting signaling [48].	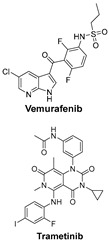
The **HER2 signaling pathway**, often overexpressed in breast cancer, promotes cell proliferation. Targeted HER2 inhibitors block this signaling, thereby reducing tumor growth.	Trastuzumab is a monoclonal antibody that targets the extracellular domain of HER2. It is widely used to treat HER2-positive breast cancer and has significantly improved outcomes for patients with this aggressive subtype [49].	Trastuzumab is a monoclonal antibody (148 kDa) of the IgG1 subclass, it consists of two heavy chains (~50 kDa each) and two light chains (~25 kDa each).
***The Epidermal Growth Factor Receptor (EGFR) signaling pathway*** promotes cell proliferation, migration, and survival. Its inhibition can effectively suppress the growth of various cancers.	**Cetuximab** is a chimeric monoclonal antibody (IgG1) that targets EGFR and is primarily used to treat colorectal cancer and head and neck squamous cell carcinoma (HNSCC) [50].	Cetuximab (152 kDa) is a chimeric monoclonal antibody made up of human and mouse components.
***The KIT Pathway*** is a receptor tyrosine kinase that, when mutated, leads to unregulated cell growth in **gastrointestinal stromal tumors** (GISTs).	**Imatinib** is a tyrosine kinase inhibitor that specifically targets KIT mutations, especially KIT exon 9 mutations in gastrointestinal stromal tumors (GISTs) [51].	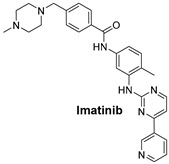
***FLT3 is a receptor tyrosine kinase*** whose mutations drive early-stage acute myeloid leukemia (AML). FLT3 inhibitors reduce leukemic cell proliferation by targeting these mutations.	**Midostaurin** is a first-generation, multi-targeted kinase inhibitor that blocks FLT3. It is primarily used in acute myeloid leukemia (AML) with FLT3 mutations and systemic mastocytosis (SM) [52].	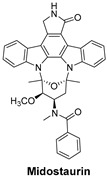
***The mechanistic target of rapamycin (mTOR) signaling pathway*** regulates cell growth, survival, and metabolism. The inhibition of mTOR can slow down cancer progression.	**Everolimus** is an mTOR inhibitor used to block the mTOR pathway, which is often dysregulated in cancers, including breast cancer [53].	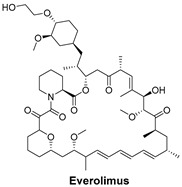
***The cyclin-dependent kinases (CDK4/6) pathway*** regulates the cell cycle and promotes cell division. Inhibiting CDK4/6 can halt cell cycle progression, leading to cancer cell death.	Ribociclib is a selective CDK4/6 inhibitor that prevents cell cycle progression from the G1 to S phase, thereby halting cancer cell proliferation and particularly effective for Triple-Positive Breast Cancer [54].	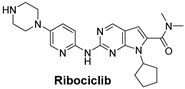
***The Poly (ADP-ribose) polymerase (PARP) pathway*** plays a key role in DNA repair. PARP inhibitors block this repair process, causing cancer cell death—especially effective in cancers with BRCA mutations.	**Niraparib** is a potent inhibitor of PARP, particularly useful in ovarian cancers, where defective DNA repair mechanisms are common [55].	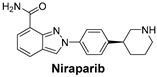
***The JAK (Janus Kinase) signaling Pathway*** regulates the immune response and hematopoiesis. In myelofibrosis, abnormal JAK activity drives excessive cell proliferation.	Ruxolitinib is a JAK1/2 inhibitor used to treat myelofibrosis by blocking the JAK–STAT signaling pathway, reducing cytokine production [56].	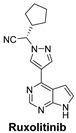

## Data Availability

All mathematical calculations and biological interpretations supporting the findings of this study are provided in the Appendix A.

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
