# Peer review of "Unlocking Cellular Memory and Gene Regulatory Networks: Pioneering the Future of Therapeutic Innovations"

_cells, 2025, doi:10.3390/cells14120903_

Round 1
Reviewer 1 Report
Comments and Suggestions for Authors
This article offers an interesting interdisciplinary perspective on the role of cellular memory in drug resistance, focusing on the role of gene regulatory networks (GRNs). It combines mathematical modeling tools with insights from recent experimental studies and proposes therapeutic strategies involving inhibitors, CRISPR tools, and synthetic biology.
However, the manuscript presents several structural, conceptual, and stylistic issues that should be addressed before it is suitable for publication.
1.- The manuscript oscillates between a review, a perspective, and a mathematical modeling paper without a well-defined narrative.
Please clearly define the purpose in the introduction—is it a review article with a theoretical proposal, or a modeling paper with experimental validation?
2.- The proposed model is based on a two-gene double positive feedback loop—a well-established motif in the literature. The analysis of noise and mutual information is interesting, but:
- The conclusions are quite predictable (higher expression → more noise → less information).
- The biological relevance is not fully contextualized.
Please justify why this model is novel or useful beyond being a didactic tool.
3.- The figures presenting numerical results (e.g., variance and MI curves) are not compared with any real experimental data.
Please include a qualitative validation of the simulations using published data (e.g., from Shaffer et al.).
4.- Please indicate why this model provides novelty beyond previous studies on positive feedback loops.
5.- Please explain if the model could be extended to more complex GRNs (more than two nodes, mixed feedback loops).
6.- Has the model been validated against real-world data (e.g., quantitative scRNA-seq)?
7.- Could your model be adapted for a case study? If yes, what real biological parameter values (e.g., for k, μ, K) could be used to adapt the model to a case study?
8.- What are the limitations of extrapolating this model to human therapy?
Please reword the abstract to better clarify the paper’s main contribution (mathematical model + therapeutic perspective).
Please better separate the theoretical modeling section from the experimental review to avoid confusion.
Include a "means and methods" section explaining the mathematical modeling approach. Its key components—such as derivations, parameter selection, simulation setup (e.g., Gillespie algorithm), and assumptions—are mentioned but not sufficiently detailed to allow reproducibility.
Please improve the explanation of the connection between theoretical equations and biological interpretation.
Comments on the Quality of English Language
The paper shows a lot of grammatical and typographical errors. A thorough review is suggested.
There are so many that I can not enumerate all of them. I will list few of them:
1.- References to bibliography are not correctly formatted along the paper. You put the number of the reference after de punctuation symbol instead of before.
2.- In the caption of figure 2, section A) you refer to the green and black lines, but there are no color lines in that graph.
3.- Sometime you do not insert a blank space between the "." signaling the end of a phrase and the beginning of the next one.
4.- Table 1 is full of typos. Please review it carefully. Sentences that end with a "," instead of a ".", text from different columns that sticks together, names of chemical structure inhibitors that are touching the line of the table, etc.
5.- No units in the graphs of figure 4.
6.- Line 323. "Shafer and et al."
7.- The phrase at line 393, 394, 395 has no verb.
Author Response
Reviewer #1
This article offers an interesting interdisciplinary perspective on the role of cellular memory in drug resistance, focusing on the role of gene regulatory networks (GRNs). It combines mathematical modeling tools with insights from recent experimental studies and proposes therapeutic strategies involving inhibitors, CRISPR tools, and synthetic biology.
However, the manuscript presents several structural, conceptual, and stylistic issues that should be addressed before it is suitable for publication.
Comment 1:The manuscript oscillates between a review, a perspective, and a mathematical modeling paper without a well-defined narrative.
Please clearly define the purpose in the introduction—is it a review article with a theoretical proposal, or a modeling paper with experimental validation?
Response 1:We are thankful to the honorable reviewer for this valuable feedback. We acknowledge that the manuscript blends elements of theoretical modeling and conceptual discussion, which may have contributed to a lack of narrative clarity. We have now revised the introduction to more clearly articulate the manuscript’s scope and purpose.
Specifically, this work is positioned as a theoretical perspective article that integrates a novel mathematical modeling framework with conceptual insights from experimental literature. While it does not include new experimental validation, it draws qualitative parallels to existing experimental findings, such as those by Shaffer et al. [Ref No.13 in main manuscript], to demonstrate the biological relevance of our model.
Our goal is to propose a computational framework that quantifies gene expression noise and mutual information in bistable regulatory networks, and to highlight its implications for cellular memory and drug resistance. The broader aim is to provide a roadmap for future experimental efforts, while offering a theoretical lens through which drug resistance mechanisms can be better understood and potentially manipulated.
We have updated the introduction accordingly to reflect this clarified structure and to guide the reader more effectively through the narrative.
All the changes in the revised main manuscript are highlighted in redfront colour.
[Ref. 13]Shaffer, S. M., et al. (2017). Rare cell variability and drug-induced reprogramming as a mode of cancer drug resistance. Nature, 546(7658), 431–435. https://doi.org/10.1038/nature22794
Comment 2:The proposed model is based on a two-gene double positive feedback loop—a well-established motif in the literature. The analysis of noise and mutual information is interesting, but:
1.The conclusions are quite predictable (higher expression → more noise → less information).
2.The biological relevance is not fully contextualized.
Please justify why this model is novel or useful beyond being a didactic tool.
Response 2:We are thankful to the esteemed reviewer for this insightful comment. We acknowledge that the double positive feedback loop is a well-characterized and extensively studied regulatory motif. However, the novelty of our work lies in integrating this classical motif with an information-theoretical framework to offer a quantitative and mechanistic explanation of how gene expression noise impacts mutual information and, consequently, cellular memory, particularly in the context of drug resistance.
While the general notion that higher expression levels can lead to increased noise is known, our work goes beyond this intuitive expectation by:
- Deriving analytical expressions for variance, covariance, and mutual information using stochastic modeling and Lyapunov analysis.
- Quantifying the degradation of mutual information as a function of gene expression strength—something that is rarely measured explicitly in models of bistability.
- Bridging the gap between information theory and gene regulatory dynamics, thereby offering a rigorous framework to explore how feedback and stochasticity influence the stability of epigenetic states.
In terms of biological relevance, we have directly contextualized our model by relating it to experimental findings from Shaffer et al., where transitions between drug-resistant and drug-susceptible states in melanoma were traced using lineage barcoding and scRNA-seq. Our model offers a theoretical rationale for the observed reversion of resistant cells to a susceptible state, which was previously attributed primarily to non-genetic heterogeneity. By showing how elevated expression levels, while initially stabilizing resistance, eventually lead to noise-induced destabilization of memory, we provide a novel explanatory mechanism that connects feedback strength, information decay, and phenotypic plasticity in drug resistance.
Finally, beyond serving as a didactic model, we believe our framework offers predictive value: it suggests that interventions aimed at increasing gene expression noise or disrupting mutual information between key regulators could destabilize drug-resistant states, offering potential strategies for therapeutic re-sensitization. This translational perspective—linking mathematical modeling to therapeutic innovation—extends the relevance of our model beyond theoretical interest.
Comment 3:The figures presenting numerical results (e.g., variance and MI curves) are not compared with any real experimental data.
Please include a qualitative validation of the simulations using published data (e.g., from Shaffer et al.).
Response 3:We appreciate this insightful comment. Our current study is primarily theoretical and was developed to provide a mathematical and information-theoretical framework for exploring how noise and feedback affect cellular memory and drug resistance. While we recognize the importance of experimental validation, our current institutional infrastructure does not yet support such advanced single-cell experiments.
That said, one of the long-term goals of our research group is to experimentally validate these findings through gene circuit engineering, time-lapse microscopy, and single-cell transcriptomics. The results presented in this work are intended to guide future experiments by predicting how gene expression noise and mutual information behave under bistable feedback regulation. In this way, our model offers a testable hypothesis for studies such as those by Shaffer et al. [Ref No.13 in main manuscript], where stochastic gene expression and state switching were observed in drug-resistant melanoma cells.
We have revised the conclusion section to clarify that while direct experimental comparison is not yet feasible, our simulations serve as a theoretical foundation that can inspire future experimental designs to explore feedback-driven memory loss and drug resistance transitions.
All the changes in the revised main manuscript are highlighted in red front colour.
Comment 4: Please indicate why this model provides novelty beyond previous studies on positive feedback loops.
Response 4:We thank the reviewer for this insightful question. While the double positive feedback loop is indeed a well-established motif in gene regulation, the novelty of our model lies in its explicit integration of mutual information theory with noise dynamics to quantitatively assess the stability of cellular memory in the context of drug resistance.
Most prior studies have focused on bistability and switch-like behavior in feedback loops [Ref. 49, 50], emphasizing deterministic or stochastic state transitions. In contrast, our work introduces a quantitative information-theoretic perspective, linking mutual information directly to the stability and fidelity of memory states. Specifically:
- Mutual Information as a Diagnostic Tool: We derive closed-form expressions for mutual information under noise and use it as a diagnostic to assess how gene-gene communication deteriorates as expression noise increases—an aspect largely unexplored in feedback loop models [Ref 51, 52].
- Bridging Theory and Drug Resistance Phenotypes: We extend the model’s implications to drug resistance in cancer, proposing that loss of mutual information can destabilize resistant states, offering a mechanistic rationale for drug re-sensitization observed in fluctuating single-cell expression patterns [Ref 13, 53].
- Analytical Link Between Noise, Feedback, and Information Loss: Unlike previous models that either simulate bistability or describe transcriptional bursting qualitatively, our approach provides analytical insight into how feedback strength and gene expression levels quantitatively affect the noise–information balance [Ref. 55].
In essence, our model extends classical positive feedback theory by embedding it within an information-processing framework, opening new avenues to analyze the fidelity of cellular decision-making and its disruption under pathological conditions.
[Ref. 49]. Gardner, T. S., Cantor, C. R. & Collins, J. J. Construction of a genetic toggle switch in Escherichia coli. Nature403, 339-342 (2000).
[Ref.50]. Ferrell Jr, J. E. Self-perpetuating states in signal transduction: positive feedback, double-negative feedback and bistability. Current opinion in cell biology14, 140-148 (2002).
[Ref.51]. Tkačik, G. & Walczak, A. M. Information transmission in genetic regulatory networks: a review. Journal of Physics: Condensed Matter23, 153102 (2011).
[Ref.52]. Ziv, E., Nemenman, I. & Wiggins, C. H. Optimal signal processing in small stochastic biochemical networks. PloS one2, e1077 (2007).
[Ref.53]. Brock, A., Chang, H. & Huang, S. Non-genetic heterogeneity—a mutation-independent driving force for the somatic evolution of tumours. Nature Reviews Genetics10, 336-342 (2009).
[Ref.55]. Bialek, W. Biophysics: searching for principles. (Princeton University Press, 2012).
All the changes in the revised main manuscript are highlighted in red front colour.
Comment 5:Please explain if the model could be extended to more complex GRNs (more than two nodes, mixed feedback loops).
Response 5:Thank you for this insightful comment. While our current model focuses on a two-node double positive feedback loop for conceptual clarity, it is readily extendable to more complex gene regulatory networks (GRNs) involving additional nodes and mixed feedback motifs. Many GRNs contain diverse topologies—such as autoregulatory, double positive, and double negative loops—known to support bistable or multistable behavior [Ref. 17]. Our framework, based on stochastic differential equations and information-theoretic measures (e.g., variance, covariance, mutual information), remains applicable as network complexity increases. It can incorporate mixed feedback architectures and higher-order interactions through expanded Hill-type equations, allowing the study of oscillatory and combinatorial regulation. Noise propagation and information flow in these extended networks can still be analyzed using generalized Lyapunov equations and multivariate covariance matrices, as established in prior work [Ref.64. We have added a note in the revised Discussion section to highlight the model’s extensibility and our intent to address these complexities in future studies.
[Ref.17] De, Nayan, et al. "Bi‐stability of the master gene regulatory network of the common dendritic precursor cell: Implications for cell differentiation." IUBMB life 72.10 (2020): 2225-2232.
[Ref.64] Paulsson, J. (2004). Summing up the noise in gene networks. Nature, 427(6973), 415–418. https://doi.org/10.1038/nature02257
Comment 6:Has the model been validated against real-world data (e.g., quantitative scRNA-seq)?
Response 6:We are thankful to the honorable reviewer for thoughtful comment. While our current study does not include direct validation against experimental data such as single-cell RNA sequencing (scRNA-seq), our model is grounded in theoretical principles that qualitatively align with experimental findings. Notably, we contextualize our results with the work of Shaffer et al. [Ref.13], who used scRNA-seq and lineage tracing to reveal dynamic transitions between drug-resistant and drug-sensitive melanoma cells. Due to current limitations in infrastructure and access to high-throughput platforms, we were unable to perform quantitative benchmarking in this study. We envision this framework as a foundation for future integration with single-cell time-course datasets, offering a valuable tool for researchers to further investigate cellular memory and drug resistance mechanisms.
Comment 7:Could your model be adapted for a case study? If yes, what real biological parameter values (e.g., for k, μ, K) could be used to adapt the model to a case study?
Response 7:We sincerely thank the reviewer for this valuable question. Our model is indeed adaptable to specific case studies through the incorporation of experimentally derived parameter values. For example, in the context of melanoma drug resistance, genes such as AXL and EGFR—identified by Shaffer et al. (2017) [Ref.13, 14]—are known to drive non-genetic variability and resistance. By modeling these genes within our double positive feedback loop framework, we can simulate the bistable behavior observed between drug-sensitive and drug-resistant states. Key parameters can be estimated as follows: synthesis rates (k) from transcriptional profiling (e.g., RNA-seq or qPCR); degradation rates (μ) from protein half-life measurements (e.g., cycloheximide or pulse-chase assays); and Hill coefficients (n) and dissociation constants (K) from binding data obtained through ChIP assays. Incorporating such parameters enables the model to reflect real biological dynamics and to quantitatively predict phenotypic transitions. Moreover, this modeling strategy is not limited to melanoma—it can be extended to other biological systems involving double positive feedback, such as cell cycle regulation or differentiation pathways, by adjusting the parameters accordingly. Thus, our framework offers a flexible and biologically grounded tool for investigating gene regulatory dynamics in diverse cellular contexts.
[Ref.14] Schuh, L. et al. Gene networks with transcriptional bursting recapitulate rare transient coordinated high expression states in cancer. Cell systems10, 363-378. e312 (2020).
Comment 8: What are the limitations of extrapolating this model to human therapy?
Response 8:Thank you for raising this important point. While our model offers valuable theoretical insights into the role of feedback loops and gene expression noise in regulating cellular memory and drug resistance, we acknowledge several limitations when extrapolating to human therapy. First, the two-gene double positive feedback loop represents a simplified abstraction of the highly complex gene regulatory networks (GRNs) in human cells, which involve numerous genes and signaling pathways [Ref 59]. Additionally, our use of additive Gaussian noise facilitates tractability but does not fully capture the bursty and non-Gaussian nature of gene expression noise influenced by chromatin state and transcriptional dynamics [Ref 69]. Accurate estimation of model parameters, such as synthesis and degradation rates, is also challenging in human systems due to biological variability and context dependence [Ref 71]. Moreover, the current model does not account for spatial heterogeneity or dynamic tumor microenvironmental effects, both of which are critical in therapeutic responses [6]. Finally, cancer progression involves extensive genetic and phenotypic heterogeneity, along with evolutionary adaptation under drug pressure—features not yet integrated into our framework [7–8]. Despite these limitations, we view our model as a foundational step that can guide experimental investigations and be extended in future studies through empirical calibration and incorporation of additional biological complexity.
[Ref.69] Alon, U. An introduction to systems biology: design principles of biological circuits. (Chapman and Hall/CRC, 2019).
[Ref.71] Dar, R. D. et al. Transcriptional burst frequency and burst size are equally modulated across the human genome. Proceedings of the National Academy of Sciences 109, 17454-17459 (2012).
Other comments 9:
9.1. Please reward the abstract to better clarify the paper’s main contribution (mathematical model + therapeutic perspective).
Response:Thank you for the thoughtful comments and suggestions regarding the clarity and focus of the abstract. Based on the feedback, we have revised the abstract to better highlight the main contributions of the manuscript, emphasizing both theoretical modeling and its translational implications.
All the changes in the revised main manuscript are highlighted in red front colour.
9.2. Please better separate the theoretical modeling section from the experimental review to avoid confusion.
Response:We are thankful to the esteemed reviewer for the suggestion and have separated the said part.
All the changes in the revised main manuscript are highlighted in red front colour.
9.3. Include a "means and methods" section explaining the mathematical modeling approach. Its key components—such as derivations, parameter selection, simulation setup (e.g., Gillespie algorithm), and assumptions—are mentioned but not sufficiently detailed to allow reproducibility.
Response: We are thankful to the esteemed reviewer for the suggestion,and it is included in the supplementary information.
9.4. Please improve the explanation of the connection between theoretical equations and biological interpretation.
Response: The explanation has provided the detailed explanation in the supplementary Information.
9.5. The paper shows a lot of grammatical and typographical errors. A thorough review is suggested.
There are so many that I cannot enumerate all of them. I will list a few of them:
9.5.1.- References to bibliography are not correctly formatted along the paper. You put the number of the reference after de punctuation symbol instead of before.
Response:We are thankful to the esteemed reviewer for the suggestion and have corrected accordingly.
9.5.2.- In the caption of figure 2, section A) you refer to the green and black lines, but there are no color lines in that graph.
Response: We have modified the figure legends where the lower part of sigmoidal curve was represented by green color (drug-susceptible state) and upper part of sigmoidal curve was represented by red color (drug-resistant state).
All the changes in the revised main manuscript are highlighted in red front colour.
9.5.3.- Sometimes you do not insert a blank space between the "." signaling the end of a phrase and the beginning of the next one.
Response: We are thankful to the reviewer for noticing this. We have corrected accordingly.
9.5.4.- Table 1 is full of typos. Please review it carefully. Sentences that end with a "," instead of a ".", text from different columns that sticks together, names of chemical structure inhibitors that are touching the line of the table, etc.
Response: We are thankful to the reviewer for noticing this. We have corrected accordingly.
9.5.5.- No units in the graphs of figure 4.
Response: We are thankful to the reviewer for noticing this. We have corrected Figure 4 accordingly.
9.5.6.- Line 323. "Shafer and et al."
Response: We are thankful to the reviewer for noticing this. We have corrected accordingly.
9.5.7.- The phrase at line 393, 394, 395 has no verb.
Response:We are thankful to the reviewer for noticing this. We have corrected Figure 4 accordingly.
Reviewer 2 Report
Comments and Suggestions for Authors
The article proposes a novel mathematical model to quantify gene expression noise and mutual information among genes involved in state transitions, offering a new perspective and methodology for studying the relationship between cellular memory and gene regulatory networks.
1. Although the article proposes a mathematical model to describe gene expression noise and mutual information, it lacks sufficient validation of the model. The authors are advised to directly verify the model's predictions through experimental means, such as designing experiments to measure the noise levels and mutual information of specific gene expressions and comparing them with model predictions.
2. The article mainly focuses on the role of cellular memory in short-term cell state transitions, with less study on the long-term stability of cellular memory over multiple cell divisions. The authors are advised to further investigate the long-term dynamic changes of cellular memory and its regulatory mechanisms.
3. Formatting of references needs to be corrected.
4. Figure 4(c) has an incorrect spelling of the vertical axis, “coariance” needs to be corrected as “covariance”
Author Response
Reviewer 2
The article proposes a novel mathematical model to quantify gene expression noise and mutual information among genes involved in state transitions, offering a new perspective and methodology for studying the relationship between cellular memory and gene regulatory networks.
Comment 1: Although the article proposes a mathematical model to describe gene expression noise and mutual information, it lacks sufficient validation of the model. The authors are advised to directly verify the model's predictions through experimental means, such as designing experiments to measure the noise levels and mutual information of specific gene expressions and comparing them with model predictions.
Response 1:We sincerely appreciate the reviewer’s valuable suggestion regarding experimental validation. Our current work focuses on establishing a rigorous theoretical framework to quantify gene expression noise and mutual information in gene regulatory networks. Although direct experimental validation was not feasible due to infrastructure limitations, we fully recognize its importance. Our model provides a foundation for designing targeted experiments to measure noise levels and mutual information in gene expression. In future work, we plan to collaborate with experimental groups to validate and refine our predictions using advanced single-cell and time-resolved techniques, thereby enhancing the biological relevance and translational impact of our findings.
Comment 2: The article mainly focuses on the role of cellular memory in short-term cell state transitions, with less study on the long-term stability of cellular memory over multiple cell divisions. The authors are advised to further investigate the long-term dynamic changes of cellular memory and its regulatory mechanisms.
Response 2:We sincerely appreciate the reviewer’s insightful comment on the long-term stability of cellular memory across multiple cell divisions. While our current model focuses on short-term cell state transitions driven by noise and feedback in gene regulatory networks, we acknowledge that long-term memory involves additional regulatory layers such as epigenetic modifications and heritable chromatin states. Incorporating factors like DNA methylation and histone modifications could deepen understanding of cellular memory’s durability and plasticity. Future work will extend our framework to include these mechanisms and validate predictions with lineage-tracing and time-series single-cell experiments. This approach aligns with recent advances in modeling epigenetic landscapes and will enhance applicability to complex biological processes including development, aging, and chronic diseases.
Comment 3: Formatting of references needs to be corrected.
Response 3: We are thankful to the honorable reviewer for noticing this. We have corrected accordingly.
Comment 4: Figure 4(c) has an incorrect spelling of the vertical axis, “coariance” needs to be corrected as “covariance.”
Response 4:We are thankful to the honorable reviewer for noticing this. We have corrected accordingly.
Reviewer 3 Report
Comments and Suggestions for Authors
- Ambiguous Scope of Model Applicability: The information-theoretic model demonstrates strong performance in small-scale GRNs, but its computational efficiency and stability in large networks (e.g., tumor-related GRNs with hundreds of nodes) remain unaddressed. Consider integrating causal inference frameworks (e.g., GFlowNet) to enhance causal interpretability of GRN dynamics.
- Disconnect Between Noise Quantification and Biological Mechanisms: Gene expression noise is treated as stochastic variables without explicit linkage to established biological mechanisms (e.g., chromatin remodeling, transcription factor competition). Incorporate epigenetic data (e.g., methylation, histone modification) to improve biological relevance.
- Limited Scope of Experimental Models: Validation relies heavily on drug resistance data (Shaffer et al.), but generalizability to other pathologies (e.g., immune evasion or metabolic disorders) is unclear. Single-cell RNA-seq captures static snapshots; temporal resolution (e.g., time-series scRNA-seq) is needed to validate dynamic transitions.
- Incomplete Multi-Omics Integration: Protein interaction and metabolomics data are notably absent, despite claims of multi-omics validation. For example, PD-L1 membrane expression dynamics (as in) could enrich mutual information analysis.
- Unaddressed Risks of CRISPR Interventions: CRISPR-based strategies for memory disruption lack critical evaluation of off-target effects or long-term safety (e.g., endogenous retrovirus reactivation). Propose synthetic biology tools (e.g., optogenetic switches) to balance precision and global regulation.
- Neglect of Immune Microenvironment Interactions: Cellular memory mechanisms likely modulate immune tolerance (e.g., T-cell exhaustion in ). Combining checkpoint inhibitors with GRN reprogramming could enhance therapeutic efficacy.
- Reproducibility Concerns: Critical formula derivations (e.g., mutual information calculation) are incompletely documented, hindering replication25. Provide supplemental appendices or open-source code.
- Suboptimal Visualization: GRN diagrams fail to highlight hub genes (e.g., drug resistance drivers). Adopt protein interaction visualization strategies from.
- Overly Simplistic Steady-State Assumptions: Cellular memory often involves non-equilibrium dynamics (e.g., cytokine bursts in). Incorporate time-dependent parameters to refine the model.
Author Response
Reviewer #3
Comment 1: Ambiguous Scope of Model Applicability: The information-theoretic model demonstrates strong performance in small-scale GRNs, but its computational efficiency and stability in large networks (e.g., tumor-related GRNs with hundreds of nodes) remain unaddressed. Consider integrating causal inference frameworks (e.g.,GFlowNet) to enhance causal interpretability of GRN dynamics.
Response 1:We thank the reviewer for this important comment regarding the scalability and broader applicability of our model. While our current focus is on a two-node double positive feedback loop for conceptual clarity, such motifs are biologically meaningful and often embedded within larger GRNs—such as the GATA1–PU.1 circuit in hematopoiesis or the MYC–PTEN–p53 axis in tumorigenesis. These sub-networks tend to retain their functional dynamics even within complex systems. Our model is inherently modular and can be extended to larger GRNs with appropriate computational adaptations. We acknowledge that applying it to large-scale networks will require improvements in computational efficiency and stability. As suggested, integrating causal inference approaches—such as Generative Flow Networks (GFlowNets)—holds promise for enhancing causal interpretability and scalability. We have incorporated this perspective in the revised manuscript to highlight future directions aimed at expanding the model’s utility for analyzing high-dimensional biological systems.
Comment 2: Disconnect Between Noise Quantification and Biological Mechanisms: Gene expression noise is treated as stochastic variables without explicit linkage to established biological mechanisms (e.g., chromatin remodeling, transcription factor competition). Incorporate epigenetic data (e.g., methylation, histone modification) to improve biological relevance.
Response 2:We sincerely thank the reviewer for highlighting the importance of linking gene expression noise to underlying biological mechanisms. Our current model serves as a foundational framework to quantify how stochastic fluctuations influence mutual information and memory stability within gene regulatory networks. While we initially modeled noise in a generalized stochastic sense, we fully acknowledge that gene expression variability is driven by specific biological processes such as chromatin remodeling, transcription factor competition, nucleosome positioning, and epigenetic modifications. To maintain tractability, we focused on noise arising from transcriptional regulation; however, we recognize that epigenetic layers—such as DNA methylation and histone modifications—play a crucial role in modulating transcriptional noise and cellular memory. In the revised manuscript, we have expanded the discussion to acknowledge how epigenetic mechanisms could be incorporated into future iterations of the model, for instance by linking them to noise parameters like burst size and frequency. We also propose the integration of epigenomic data (e.g., ATAC-seq, ChIP-seq) alongside scRNA-seq as a promising direction for refining the model’s biological fidelity. This connection will be critical in extending the utility of our framework to more mechanistically grounded studies of gene regulation.
Comment 3: Limited Scope of Experimental Models: Validation relies heavily on drug resistance data, but generalizability to other pathologies (e.g., immune evasion or metabolic disorders) is unclear. Single-cell RNA-seq captures static snapshots; temporal resolution (e.g., time-series scRNA-seq) is needed to validate dynamic transitions.
Response 3:We thank the reviewer for this thoughtful comment on the scope and validation of our model. While our current study emphasizes drug resistance due to the availability of high-resolution single-cell data in this domain, the core principles of our model—such as gene expression noise and feedback regulation—are broadly applicable to other pathologies involving dynamic cell-state transitions. For example, immune evasion mechanisms (e.g., T-cell exhaustion) and metabolic reprogramming in cancer and chronic inflammation also exhibit memory-like behavior that can be modeled within our framework.
We agree that standard single-cell RNA-seq provides static snapshots, limiting dynamic validation. To address this, we propose the integration of emerging time-resolved transcriptomic techniques—such as RNA velocity, scSLAM-seq, and scNT-seq – which can infer or directly measure transcriptional dynamics at single-cell resolution. These methods offer a powerful way to validate and refine our model's predictions over time.
We believe that these additions will not only address the reviewer’s concerns but also pave the way for broader applications of our model in understanding complex biological processes.
Comment 4: Incomplete Multi-Omics Integration: Protein interaction and metabolomics data are notably absent, despite claims of multi-omics validation. For example, PD-L1 membrane expression dynamics could enrich mutual information analysis.
Response 4:We thank the reviewer for this valuable comment. In our study, the term multi-omics integration primarily refers to transcriptomic data (e.g., scRNA-seq) and chromatin accessibility (e.g., ATAC-seq), as our model focuses on transcriptional regulation and its role in cellular memory. While we have not incorporated proteomic or metabolomic data in the current framework, we agree that integrating such layers—particularly protein-level dynamics like PD-L1 membrane expression—could enhance the biological depth of mutual information analysis.
We acknowledge this limitation and now clarify this scope in the revised manuscript. We also highlight that future extensions of the model could incorporate proteomic and metabolomic data to capture post-transcriptional and metabolic influences on cell-state dynamics, especially in immune contexts such as PD-L1 regulation.
Comment 5: Unaddressed Risks of CRISPR Interventions: CRISPR-based strategies for memory disruption lack critical evaluation of off-target effects or long-term safety (e.g., endogenous retrovirus reactivation). Propose synthetic biology tools (e.g., optogenetic switches) to balance precision and global regulation.
Response 5:We thank the reviewer for highlighting the critical issue of off-target effects and long-term safety risks associated with CRISPR-based memory disruption. To address these concerns, we suggest the integration of synthetic biology tools such as Synthetic Transcriptional Interference Networks (STINs) and optogenetic switches. STINs employ programmable RNA molecules (e.g., synthetic or antisense RNAs) to transiently suppress resistance gene expression without genomic alteration, reducing off-target risks and enabling reversible control. Additionally, optogenetic switches allow precise spatiotemporal regulation of CRISPR activity using light, minimizing unintended gene activation and improving safety. These approaches collectively enable precise CRISPR interventions particularly in unaddressed Risks of CRISPR Interventions.
Comment 6: Neglect of Immune Microenvironment Interactions: Cellular memory mechanisms likely modulate immune tolerance (e.g., T-cell exhaustion). Combining checkpoint inhibitors with GRN reprogramming could enhance therapeutic efficacy.
Response 6: We thank the reviewer for emphasizing the important role of the immune microenvironment in cellular memory and therapeutic response. We agree that immune tolerance mechanisms, such as T-cell exhaustion, are closely linked to gene regulatory networks (GRNs) controlling cellular states. Integrating immune interactions into our model is a promising future direction to enhance biological relevance and therapeutic impact. Recent studies show that reprogramming GRNs in exhausted T cells, especially alongside checkpoint inhibitors, can restore immune function and improve anti-tumor efficacy. While our current focus is on tumor cell-intrinsic feedback loops, we plan to extend the model in the future to incorporate immune-tumor cross-talk and multi-cellular dynamics.
Comment 7: Reproducibility Concerns: Critical formula derivations (e.g., mutual information calculation) are incompletely documented, hindering replication. Provide supplemental appendices or open-source code.
Response 7:We thank the reviewer for emphasizing the importance of reproducibility. To address this, we have added a comprehensive supplementary document that includes detailed derivations and step-by-step explanations of key analytical formulas, including the mutual information calculations. We believe this will support transparency, facilitate replication, and enable further research by interested readers.
Comment 8: Suboptimal Visualization: GRN diagrams fail to highlight hub genes (e.g., drug resistance drivers). Adopt protein interaction visualization strategies.
Response 8:We appreciate the reviewer’s observation. In this study, we focused specifically on the transcriptional regulation aspect of drug resistance, emphasizing the role of master transcription factors within gene regulatory networks (GRNs). As our model centers on transcription-level dynamics, we did not incorporate protein interaction networks. However, we acknowledge that integrating protein-level interactions and visualizing hub genes could further enhance interpretability, and we will consider this in future model extensions.
Comment 9: Overly Simplistic Steady-State Assumptions: Cellular memory often involves non-equilibrium dynamics (e.g., cytokine bursts). Incorporate time-dependent parameters to refine the model.
Response 9:We thank the reviewer for the insightful comment. We acknowledge that cellular memory often arises from non-equilibrium dynamics, such as transient cytokine bursts and fluctuating signaling pathways. While our current model adopts steady-state assumptions to provide foundational insights into gene expression noise and mutual information, we agree that incorporating time-dependent parameters would improve its biological realism. In future work, we plan to extend the framework using stochastic time-dependent models, such as delay differential equations or variable rate constants, to better capture transient gene expression responses and dynamic memory behaviors.
Reviewer 4 Report
Comments and Suggestions for Authors
The authors summarized the relationship between cellular memory and gene regulatory networks, as well as potential therapeutic strategies for drug resistance controlling gene and superior drug efficacy. The organization of this perspective is very clear and layout is easy to read. There are a few suggestions that the authors can make to improve this manuscript.
- In Table 1, the text is too verbose and it could be better summarized and more concise.
- The format of figures are not consistent, for example: font, location of subtitles...
- In References section, there are repeating numbering.
Author Response
Reviewer #4
The authors summarized the relationship between cellular memory and gene regulatory networks, as well as potential therapeutic strategies for drug resistance controlling gene and superior drug efficacy. The organization of this perspective is very clear, and the layout is easy to read. There are a few suggestions that the authors can make to improve this manuscript.
Comments 1. In Table 1, the text is too verbose, and it could be better summarized and more concise.
Response 1: We are thankful to the honorable reviewer for noticing this. We have corrected accordingly.
All the changes in the revised main manuscript are highlighted in red front colour.
Comments 2. The format of figures is not consistent, for example: font, location of subtitles...
Response 2: We are thankful to the honorable reviewer for noticing this. We have corrected accordingly.
In the References section, there are repeated numbering.
Response 3: We are thankful to the honorable reviewer for noticing this. We have corrected accordingly.
Reviewer 5 Report
Comments and Suggestions for Authors
Regarding the Manuscript ID: cells-3643000, Title “Unlocking Cellular Memory and Gene Regulatory Networks: Pioneering the Future of Therapeutic Innovations”
The manuscript is very interesting by the initiative mathematical analysis on the feedback loop for gene transcription and its relationship with gene regulatory networks to explain the cell´s memory of a cell. However, it is necessary to consider several aspects to improve this manuscript.
- The analysis considers mainly two genes with mutual feedback which keep the cell’s memory. This analysis is correct if such a cell’s memory depends on two related genes. Also consider the “noise effect which can be extrinsic source (variation on cell’s components like transcription factors and intrinsic source derived from stochastic nature of transcription and translation processes. However, did not consider in this mathematical analysis the transcription regulation given by miRNAs, and other small RNAs, attenuation on translation, mRNA half time depending on his mRNA structure, also, if one gene regulates the transcription of several genes and some of them offer positive feedback or not. Furthermore, there are gene expressions in a physiological moment where several genes participate (multigene process).
- Other considerations consist of the effect of environmental influencing factors that change the epigenetic of a genome influencing a change cell’s behavior which could favor a particular phenotype.
- On the other hand, the authors consider punctual mutation that can direct by possible therapy, for example cancer (sensible or resistant to drug therapy), but curiously, cancer shows a diversity of chromosomal changes which can generate not only one resistant phenotype and then it must be considered.
- Then the manuscript must be proposed initially to analyze simple mutation which can break the stable state of the cell’s memory and more complex mathematical analysis must be done
- It is interesting that only two works of cell memory are analyzed, although five references are related to this topic. The references are not well referenced in the text (i.e. Saffer et al in line 324 and references inserted are 15,63 in line 329. Then must be reviewed that all text corresponds to the proper reference.
- The Shaffer’s analysis was made using cells in culture, that means the cell environment is controlled and few parameters are influencing the cell genotype and phenotype. Then the discussion must be considered this condition.
- The discussion, conclusion and therapy proposal must be more specific to the mathematical analysis
- Orthography must be reviewed to reduce small errors.
Has several orthographic errors and must be reviewed
Author Response
Reviewer #5
Regarding the Manuscript ID: cells-3643000, Title “Unlocking Cellular Memory and Gene Regulatory Networks: Pioneering the Future of Therapeutic Innovations”
The manuscript is very interesting by the initiative mathematical analysis on the feedback loop for gene transcription and its relationship with gene regulatory networks to explain the cell´s memory of a cell. However, it is necessary to consider several aspects to improve this manuscript.
Comment 1: The analysis considers mainly two genes with mutual feedback which keep the cell’s memory. This analysis is correct if such a cell’s memory depends on two related genes. Also, it considers the noise effect which can be from extrinsic sources (e.g., variation in transcription factors) and intrinsic sources (e.g., stochasticity in transcription/translation). However, the mathematical analysis does not account for transcriptional regulation by miRNAs or other small RNAs, translational attenuation, mRNA half-life determined by structure, or cases where one gene regulates multiple downstream genes, some with positive feedback. Furthermore, real physiological gene expression often involves multigene processes.
Response 1: We thank the honourable reviewer for this detailed and valuable observation. We acknowledge that our current model, focusing on a two-gene double positive feedback loop, represents a simplified but fundamental motif in gene regulatory networks (GRNs) that captures key aspects of cellular memory. While this minimal model effectively illustrates the interplay of noise and mutual information in stabilizing cell states, we agree that real cellular contexts involve additional layers of complexity such as regulation by miRNAs and other non-coding RNAs, translational control, mRNA stability, and multigene regulatory cascades.Incorporating these factors—such as miRNA-mediated repression, variable mRNA half-lives influenced by secondary structure, and multi-target regulation—would indeed enhance biological realism. These processes introduce additional sources of noise and feedback that can significantly affect gene expression dynamics and cellular memory. Extending the model to capture multigene networks with mixed feedback loops and post-transcriptional regulation is an important future direction, as it will allow us to better represent physiological gene expression in complex systems.
Our current work establishes a foundational theoretical framework that can be expanded to include these layers. Integrating such mechanisms would likely require more sophisticated mathematical and computational approaches, including stochastic simulations and multi-omics data integration, which we plan to pursue in follow-up studies.
Comment 2: Another consideration is the effect of environmental factors that induce epigenetic changes in the genome, potentially altering cell behavior and favoring specific phenotypes.
Response 2:We appreciate the reviewer’s insightful comment on how environmental factors induce epigenetic changes that influence cellular behavior and phenotype stability. Epigenetic modifications such as DNA methylation, histone changes, and chromatin remodeling critically modulate gene expression and cellular memory. While our current model focuses on gene regulatory network dynamics and transcriptional noise, we acknowledge that integrating epigenetic regulation and environmental stimuli is key to capturing the full complexity of cellular behavior. These influences can alter gene expression parameters, affecting noise and mutual information. Our future work will aim to incorporate epigenetic datasets and signaling pathways to better capture these complex interactions and their role in cellular memory and phenotypic plasticity.
Comment 3: The manuscript focuses on point mutations as potential therapeutic targets (e.g., in cancer drug sensitivity/resistance). However, cancer typically involves diverse chromosomal alterations, which may produce multiple resistant phenotypes. This complexity should be addressed.
Response 3:We thank the reviewer for emphasizing the complexity of cancer genetics beyond point mutations. Indeed, chromosomal alterations such as copy number variations, translocations, and aneuploidy contribute significantly to tumor heterogeneity and multiple resistant phenotypes. Our current model focuses on simplified gene regulatory feedback loops to elucidate fundamental principles of cellular memory and noise effects. We acknowledge that fully capturing cancer complexity requires models integrating diverse genetic and epigenetic alterations and their network interactions. We recognize this limitation and plan to develop expanded frameworks addressing these aspects in future work.
Comment 4: The manuscript should begin by analyzing simple mutations that can disrupt the stability of cellular memory and then progress toward more complex mathematical models.
Response 4:We appreciate the reviewer’s suggestion. However, our model focuses on cell state dynamics driven by gene regulatory activity and non-genetic heterogeneity, rather than mutation-dependent cellular memory. The emergence of drug-resistant cell populations in our framework results from gene expression variability and feedback, not from genetic mutations. Therefore, we have intentionally omitted analysis of simple mutations disrupting cellular memory stability, as it falls outside the scope of this study.
Comment 5: Only two works on cellular memory are analyzed, though five references are cited. The referencing needs to be reviewed carefully (e.g., Shaffer et al. is mentioned in line 324, while references 15 and 63 are cited in line 329). All text citations should accurately correspond to their references.
Response 5:We thank the reviewer for pointing out the inconsistencies in referencing. We have carefully reviewed and corrected all citations to ensure that each in-text reference accurately corresponds to the appropriate source in the reference list. Additionally, we have expanded the discussion to more thoroughly incorporate all five relevant works on cellular memory, providing clearer context and integration of these studies throughout the manuscript.
Comment 6: Shaffer’s analysis used cultured cells, where the environment is controlled, and relatively few parameters influence genotype/phenotype. This experimental condition should be discussed and acknowledged.
Response 6:We appreciate the reviewer’s insight regarding the controlled environment of cultured cells in the referenced experimental studies. Indeed, in vitro conditions simplify many environmental factors present in vivo—such as immune interactions, extracellular matrix effects, and systemic influences—that impact gene expression and cellular phenotypes. We have acknowledged this limitation noting that while cultured cell studies offer valuable mechanistic insights, translating findings to complex physiological contexts requires further validation. Future work will focus on incorporating more physiologically relevant models, such as patient-derived xenografts and organoids, alongside microenvironmental factors to better capture cellular memory and drug resistance dynamics in vivo. Thank you for highlighting this important consideration, which enhances the translational relevance of our study.
Comment7: The discussion, conclusion, and therapeutic proposals must be more closely tied to the mathematical analysis presented in the manuscript.
Response 7:We thank the reviewer for the suggestion. We have revised the manuscript to better connect the mathematical analysis with the discussion, conclusion, and therapeutic proposals, clearly showing how our model informs biological insights and treatment strategies.
All the changes in the revised main manuscript are highlighted in red front colour.
Comment 8: Orthography should be reviewed to eliminate minor grammatical and typographical errors throughout the manuscript.
Response 8:We appreciate the reviewer’s comment and have thoroughly reviewed the manuscript to correct grammatical and typographical errors to ensure clarity and consistency throughout the text.
All the changes in the revised main manuscript are highlighted in red front colour.
Round 2
Reviewer 1 Report
Comments and Suggestions for Authors
Thanks for your comments to my review.
You have improved the manuscript, although I think some of my comments that you addressed correctly in your response, deserve an amendment in the text, for example comment #2, #5, #6, #7, and #8.
I think that a Means and Methods sections should be included in the paper, and not only as supplementary information.
The formatting of table 1 was nicer in the original version. The interlining you have introduced is too big (on my humble opinion). Besides, just using horizontal lines to separate files (instead of horizontal and vertical lines) made it easer to read (again, on my humble opinion).
Author Response
Reviewer1:
Thanks for your comments to my review.
Comment: You have improved the manuscript, although I think some of my comments that you addressed correctly in your response, deserve an amendment in the text, for example comment #2, #5, #6, #7, and #8.
Response: We appreciate the reviewer’s comment. We have revised our manuscript accordingly and the changes are highlighted red.
Comment: I think that a Means and Methods sections should be included in the paper, and not only as supplementary information.
Response: We have included the mean and method section in the revised manuscript.
Comment: The formatting of table 1 was nicer in the original version. The interlining you have introduced is too big (on my humble opinion). Besides, just using horizontal lines to separate files (instead of horizontal and vertical lines) made it easer to read (again, on my humble opinion).
Response: In the previous version we modified our table based on the comment of other reviewer.
Reviewer 2 Report
Comments and Suggestions for Authors
All raised issues in the previous review were answered and resolved. The article can be accepted in present form.
Author Response
Reviewer2:
All raised issues in the previous review were answered and resolved. The article can be accepted in present form.
Response: Thank you very much for your positive feedback.
Reviewer 5 Report
Comments and Suggestions for Authors
The manuscript ID cells-3643000 Title “Unlocking Cellular Memory and Gene Regulatory Networks: Pioneering the Future of Therapeutic Innovations” has been improved enough following the reviewer’s comments. I do not have any more comments about it.
Author Response
Reviewer5:
The manuscript ID cells-3643000 Title “Unlocking Cellular Memory and Gene Regulatory Networks: Pioneering the Future of Therapeutic Innovations” has been improved enough following the reviewer’s comments. I do not have any more comments about it.
Response: Thank you so much for appreciating our work.